# PEWA: Patch-based Exponentially Weighted Aggregation for image denoising

**Charles Kervrann**
Inria Rennes - Bretagne Atlantique
Serpico Project-Team
Campus Universitaire de Beaulieu, 35 042 Rennes Cedex, France
`charles.kervrann@inria.fr`

## Abstract

Patch-based methods have been widely used for noise reduction in recent years. In this paper, we propose a general statistical aggregation method which combines image patches denoised with several commonly-used algorithms. We show that weakly denoised versions of the input image obtained with standard methods, can serve to compute an efficient patch-based aggregated estimator. In our approach, we evaluate the Stein's Unbiased Risk Estimator (SURE) of each denoised candidate image patch and use this information to compute the exponential weighted aggregation (EWA) estimator. The aggregation method is flexible enough to combine any standard denoising algorithm and has an interpretation with Gibbs distribution. The denoising algorithm (PEWA) is based on a MCMC sampling and is able to produce results that are comparable to the current state-of-the-art.

## 1 Introduction

Several methods have been proposed to solve the image denoising problem including anisotropic diffusion [15], frequency-based methods [26], Bayesian and Markov Random Fields methods [20], locally adaptive kernel-based methods [17] and sparse representation [10]. The objective is to estimate a clean image generally assumed to be corrupted with additive white Gaussian (AWG) noise. In recent years, state-of-the-art results have been considerably improved and the theoretical limits of denoising algorithms are currently discussed in the literature [4, 14]. The most competitive methods are mostly patch-based methods, such as BM3D [6], LSSC [16], EPLL [28], NL-Bayes [12], inspired from the N(on)L(ocal)-means [2]. In the NL-means method, each patch is replaced by a weighted mean of the most similar patches found in the noisy input image. BM3D combines clustering of noisy patches, DCT-based transform and shrinkage operation to achieve the current state-of-the-art results [6]. PLOW [5], S-PLE [24] and NL-Bayes [12], falling in the same category of the so-called *internal* methods, are able to produce very comparable results. Unlike BM3D, covariances matrices of clustered noisy patches are empirically estimated to compute a Maximum A Posteriori (MAP) or a Minimum-Mean-Squared-Error (MMSE) estimate. The aforementioned algorithms need two iterations [6, 12, 18] and the performances are surprisingly very close to the state-of-the-art in average while the motivation and the modeling frameworks are quite different. In this paper, the proposed Patch-based Exponential Weighted Aggregation (PEWA) algorithm, requiring no patch clustering, achieves also the state-of-the-art results.

A second category of patch-based *external* methods (e.g. FoE [20], EPLL [28], MLP [3]) has been also investigated. The principle is to approximate the noisy patches using a set of patches of an external learned dictionary. The statistics of a noise-free training set of image patches, serve as priors for denoising. EPLL computes a prior from a mixture of Gaussians trained with a database of clean image patches [28]; denoising is then performed by maximizing the so-called Expected Patch Log Likelihood (EPLL) criteria using an optimization algorithm. In this line of work, a multi-

layer perceptron (MLP) procedure exploiting a training set of noisy and noise-free patches was able to achieve the state-of-the-art performance [3]. Nevertheless, the training procedure is dedicated to handle a fixed noise level and the denoising method is not flexible enough, especially for real applications when the signal-to-noise ratio is not known.

Recently, the similarity of patch pairs extracted from the input noisy image and from clean patch dataset has been studied in [27]. The authors observed that more repetitions are found in the same noisy image than in a clean image patch database of natural images; also, it is not necessary to examine patches far from the current patch to find good matching. While the external methods are attractive, computation is not always feasible since a very large collection of clean patches are required to denoise all patches in the input image. Other authors have previously proposed to learn a dictionary on the noisy image [10] or to combine *internal* and *external* information (LSSC) [16]. In this paper, we focus on *internal* methods since they are more flexible for real applications than *external* methods. They are less computationally demanding and remain the most competitive.

Our approach consists in estimating an image patch from "weakly" denoised image patches in the input image. We consider the general problem of combining multiple basic estimators to achieve an estimation accuracy not much worse than that of the "best" single estimator in some sense. This problem is important for practical applications because single estimators often do not perform as well as their combinations. The most important and widely studied aggregation method that achieves the optimal average risk is the Exponential Weighted Aggregation (EWA) algorithm [13, 7, 19]. Salmon & Le Pennec have already interpreted the NL-means as a special case of the EWA procedure but the results of the extended version described in [21] were similar to [2].

Our estimator combination is then achieved through a two-step procedure, where multiple estimators are first computed and are then combined in a second separate computing step. We shall see that the proposed method can be thought as a boosting procedure [22] since the performance of the pre-computed estimators involved in the first step are rather poor, both visually and in terms of peak signal-to-noise ratio (PSNR). Our contributions are the following ones:

1. We show that "weak" denoised versions of the input noisy images can be combined to get a boosted estimator.

2. A spatial Bayesian prior and a Gibbs energy enable to select good candidate patches.

3. We propose a dedicated Monte Carlo Markov Chain (MCMC) sampling procedure to compute efficiently the PEWA estimator.

The experimental results are comparable to BM3D [6] and the method is implemented efficiently since all patches can be processed independently.

## 2  Patch-based image representation and SURE estimation

Formally, we represent a $n$-dimensional image patch at location $x \in \mathcal{X} \subset \mathbb{R}^2$ as a vector $f(x) \in \mathbb{R}^n$. We define the observation patch $v(x) \in \mathbb{R}^n$ as: $v(x) = f(x) + \varepsilon(x)$ where $\varepsilon(x) \sim \mathcal{N}(0, \sigma^2 I_{n \times n})$ represents the errors. We are interested in an estimator $\widehat{f}(x)$ of $f(x)$ assumed to be independent of $f(x)$ that achieves a small $L_2$ risk. We consider the Stein's Unbiased Risk Estimator

$$R(\widehat{f}(x)) = \|v(x) - \widehat{f}(x)\|_n^2 - n\sigma^2$$

in the Mean Square Error sense such that $\mathbb{E}[R(\widehat{f}(x))] = \mathbb{E}[\|f(x) - \widehat{f}(x)\|_n^2]$ ($\mathbb{E}$ denotes the mathematical expectation). SURE has been already investigated for image denoising using NL-means [23, 9, 22, 24] and for image deconvolution in [25].

## 3  Aggregation by exponential weights

Assume a family $\{f_\lambda(x), \lambda \in \Lambda\}$ of functions such that the mapping $\lambda \to f_\lambda(x)$ is measurable and $\Lambda = \{1, \cdots, M\}$. Functions $f_\lambda(x)$ can be viewed as some pre-computed estimators of $f(x)$ or "weak" denoisers independent of observations $v(x)$, and considered as frozen in the following. The set of $M$ estimators is assumed to be very large, that is composed of several hundreds of thousands

of candidates. In this paper, we consider aggregates that are weighted averages of the functions in the set $\{f_\lambda(x), \lambda \in \Lambda\}$ with some data-dependent weights:

$$\widehat{f}(x) \; = \; \sum_{\lambda=1}^{M} w_\lambda(x)\, f_\lambda(x) \;\; \text{such that} \;\; w_\lambda(x) \geq 0 \;\; \text{and} \;\; \sum_{\lambda=1}^{M} w_\lambda(x) = 1. \tag{1}$$

As suggested in [19], we can associate two probability measures $\boldsymbol{w}(x) = \{w_1(x), \cdots, w_M(x)\}$ and $\boldsymbol{\pi}(x) = \{\pi_1(x), \cdots, \pi_M(x)\}$ on $\{1, \cdots, M\}$ and we define the Kullback-Leibler divergence as:

$$D_{KL}(\boldsymbol{w}(x), \boldsymbol{\pi}(x)) \; = \; \sum_{\lambda=1}^{M} w_\lambda(x) \log\left(\frac{w_\lambda(x)}{\pi_\lambda(x)}\right). \tag{2}$$

The exponential weights are obtained as the solution of the following optimization problem:

$$\widehat{\boldsymbol{w}}(x) \; = \; \arg\min_{\boldsymbol{w}(x) \in \mathbb{R}^M} \left\{ \sum_{\lambda=1}^{M} w_\lambda(x)\phi(R(f_\lambda(x))) + \beta\, D_{KL}(\boldsymbol{w}(x), \boldsymbol{\pi}(x)) \right\} \text{ subject to (1)} \tag{3}$$

where $\beta > 0$ and $\phi(z)$ is a function of the following form $\phi(z) = |z|$. From the Karush-Kuhn-Tucker conditions, the unique closed-form solution is

$$w_\lambda(x) \; = \; \frac{\exp(-\phi(R(f_\lambda(x)))/\beta)\, \pi_\lambda(x)}{\sum_{\lambda'=1}^{M} \exp(-\phi(R(f_{\lambda'}(x)))/\beta)\, \pi_{\lambda'}(x)}, \tag{4}$$

where $\beta$ can be interpreted as a "temperature" parameter. This estimator satisfies oracle inequalities of the following form [7]:

$$\mathbb{E}[R(\widehat{f}(x))] \; \leq \; \min_{\boldsymbol{w}(x) \in \mathbb{R}^M} \left\{ \sum_{\lambda=1}^{M} w_\lambda(x)\phi(R(f_\lambda(x))) + \beta\, D_{KL}(\boldsymbol{w}(x), \boldsymbol{\pi}(x)) \right\}. \tag{5}$$

The role of the distribution $\boldsymbol{\pi}$ is to put a prior weight on the functions in the set. When there is no preference, the uniform prior is a common choice but other choices are possible (see [7]).

In the proposed approach, we define the set of estimators as the set of patches taken in denoised versions of the input image $v$. The next question is to develop a method to efficiently compute the sum in (1) since the collection can be very large. For a typical image of $N = 512 \times 512$ pixels, we could potentially consider $M = L \times N$ pre-computed estimators if we apply $L$ denoisers to the input image $v$.

## 4    PEWA: Patch-based EWA estimator

Suppose that we are given a large collection of $M$ competing estimators. These basis estimators can be chosen arbitrarily among the researchers favorite denoising algorithm: Gaussian, Bilateral, Wiener, Discrete Cosine Transform or other transform-based filterings. Let us emphasize here that the number of basic estimators $M$ is not expected to grow and is typically very large ($M$ is chosen on the order of several hundreds of thousands). In addition, the essential idea is that these basic estimators only slightly improve the PSNR values of a few dBs.

Let us consider $u_\ell, \ell = 1, \cdots, L$ denoised versions of $v$. A given pre-computed patch estimator $f_\lambda(x)$ is then a $n$-dimensional patch taken in the denoised image $u_\ell$ at any location $y \in \mathcal{X}$, in the spirit of the NL-means algorithm which considers only the noisy input patches for denoising. The proposed estimator is then more general since a set of denoised patches at a given location are used. Our estimator is then of the following form if we choose $\phi(z) = |z|$:

$$\widehat{f}(x) \; = \; \frac{1}{Z(x)} \sum_{\ell=1}^{L} \sum_{y \in \mathcal{X}} e^{-|R(u_\ell(y))|/\beta}\, \pi_\ell(y)\, u_\ell(y), \quad Z(x) = \sum_{\ell'=1}^{L} \sum_{y' \in \mathcal{X}} e^{-|R(u_{\ell'}(y'))|/\beta}\, \pi_{\ell'}(y) \tag{6}$$

where $Z(x)$ is a normalization constant. Instead of considering a uniform prior over the set of denoised patches taken in the whole image, it is appropriate to encourage patches located in the

neighborhood of $x$ [27]. This can be achieved by introducing a spatial Gaussian prior $G_\tau(z) \propto e^{-z^2/(2\tau^2)}$ in the definition as

$$\widehat{f}_{PEWA}(x) \quad = \quad \frac{1}{Z(x)} \sum_{\ell=1}^{L} \sum_{y \in \mathcal{X}} e^{-|R(u_\ell(y))|/\beta} \, G_\tau(x-y) \, u_\ell(y). \tag{7}$$

The Gaussian prior has a significant impact on the performance of the EWA estimator. Moreover, the practical performance of the estimator strongly relies on an appropriate choice of $\beta$. This important question has been thoroughly discussed in [13] and $\beta = 4\sigma^2$ is motivated by the authors. Finally, our patch-based EWA (PEWA) estimator can be written in terms of energies and Gibbs distributions as:

$$\widehat{f}_{PEWA}(x) \quad = \quad \frac{1}{Z(x)} \sum_{\ell=1}^{L} \sum_{y \in \mathcal{X}} e^{-E(u_\ell(y))} \, u_\ell(y), \quad Z(x) = \sum_{\ell'=1}^{L} \sum_{y' \in \mathcal{X}} e^{-E(u_{\ell'}(y'))}, \tag{8}$$

$$E(u_\ell(y)) \quad = \quad \frac{|\|v(x) - u_\ell(y)\|_n^2 - n\sigma^2|}{4\sigma^2} + \frac{\|x-y\|_2^2}{2\tau^2}.$$

The sums in (8) cannot be computed, especially when we consider a large collection of estimators. In that sense, it differs from the NL-means methods [2, 11, 23, 9] which exploits patches generally taken in a neighborhood of fixed size. Instead, we propose a Monte-Carlo sampling method to approximately compute such an EWA when the number of aggregated estimators is large [1, 19].

### 4.1 Monte-Carlo simulations for computation

Because of the high dimensionality of the problem, we need efficient computational algorithms, and therefore we suggest a stochastic approach to compute the PEWA estimator. Let us consider a random process $(F_n(x))_{n \geq 0}$ consisting in an initial noisy patch $F_0(x) = v(x)$. The proposed Monte-Carlo procedure recommended to compute the estimator is based on the following Metropolis-Hastings algorithm:

Draw a patch by considering a two-stage drawing procedure:

- draw uniformly a value $\ell$ in the set $\{1, 2, \cdots, L\}$.
- draw a pixel $y = y_c + \gamma, y \in \mathcal{X}$, with $\gamma \sim \mathcal{N}(0, I_{2 \times 2}\tau^2)$ and $y_c$ is the position of the current patch. At the initialization $y_c = x$.

Define $F_{n+1}(x)$ as: $F_{n+1}(x) = \begin{cases} u_\ell(y) & \text{if } \alpha \leq e^{-\Delta E(u_\ell(y)), F_n(x))} \\ F_n(x) & \text{otherwise} \end{cases}$ \hfill (9)

where $\alpha$ is a random variable: $\alpha \sim U[0,1]$ and $\Delta E(u_\ell(y), F_n(x)) \overset{\triangle}{=} E(u_\ell(y)) - E(F_n(x))$.

If we assume the Markov chain is ergodic, homogeneous, reductible, reversible and stationary, for any $F_0(x)$, we have almost surely

$$\lim_{T \to +\infty} \frac{1}{T - T_b} \sum_{n=T_b}^{T} F_n(x) \approx \widehat{f}_{PEWA}(x) \tag{10}$$

where $T$ is the maximum number of samples of the Monte-Carlo procedure. It is also recommended to introduce a burn-in phase to get a more satisfying estimator. Hence, the first $T_b$ samples are discarded in the average The Metropolis-Hastings rule allows reversibility and then stationarity of the Markov chain. The chain is irreducible since it is possible to reach any patch in the set of possible considered patches. The convergence is ensured when $T$ tends to infinity. In practice, $T$ is assumed to be high to get a reasonable approximation of $\widehat{f}_{PEWA}(x)$. In our implementation, we set $T \approx 1000$ and $T_b = 250$ to produce fast and satisfying results. To improve convergence speed, we can use several chains instead of only one [21].

In the Metropolis-Hastings dynamics, some patches are more frequently selected than others at a given location. The number of occurrences of a particular candidate patch can be then evaluated. In constant image areas, there is probably no preference for any one patch over any other and a low number of candidate patches is expected along image contours and discontinuities.

## 4.2 Patch overlapping and iterations

The next step is to extend the PEWA procedure at every position of the entire image. To avoid block effects at the patch boundaries, we overlap the patches. As a result, for the pixels lying in the overlapping regions, we obtain multiple EWA estimates. These competing estimates must be fused or aggregated into the single final estimate. The final aggregation can be performed by a weighted average of the multiple EWA estimates as suggested in [21, 5, 22]. The simplest method of aggregating such multiple estimates is to average them using equal weights. Such uniform averaging provided the best results in our experiments and amounts to fusing $n$ independent Markov chains.

The proposed implementation proceeds in two identical iterations. At the first iteration, the estimation is performed using several denoised versions of the noisy image. At the second iteration, the first estimator is used as an additional denoised image in the procedure to improve locally the estimation as in [6, 12]. The second iteration improves the PSNR values in the range of 0.2 to 0.5 dB as demonstrated by the experiments presented in the next section. Note that the first iteration is able to produce very satisfying results for low and medium levels of noise. In practical imaging, we use the method described in [11] to estimate the noise variance $\sigma^2$ for real-world noisy images.

## 5 Experimental results

We evaluated the PEWA algorithm on 25 natural images showing natural, man-made, indoor and outdoor scenes (see Fig. 1). Each original image was corrupted with white Gaussian noise with zero mean and variance $\sigma^2$. In our experiments, the best results are obtained with $n = 7 \times 7$ patches and $L = 4$ images $u_l$ denoised with DCT-based transform [26] ; we consider three different DCT shrinkage thresholds: $1.25\sigma, 1.5\sigma$ and $1.75\sigma$ to improve the PSNR of 1 to 6 db at most, depending on $\sigma$ and images (see Figs. 2-3). The fourth image is the noisy input image itself. We evaluated the algorithm with a larger number $L$ of denoised images and the quality drops by 0.1 db to 0.3 db, which is visually imperceptible. Increasing $L$ suggest also to considering more than 1000 samples since the space of candidate patches is larger. The prior neighborhood size corresponds to a disk of radius $\tau = 7$ pixels but it can be smaller.

Performances of PEWA and other methods are quantified in terms of PSNR values for several noise levels (see Tables 1-3). Table 1 reports the results obtained with PEWA on each individual image for different values of standard deviation of noise. Table 2 compares the average PSNR values on these 25 images obtained by PEWA (after 1 and 2 iterations) and two state-of-the-art denoising methods [6, 12]. We used the implementations provided by the authors: BM3D (http://www.cs.tut.fi/~foi/GCF-BM3D/) and NL-Bayes (www.ipol.im). The best PSNR values are in bold and the results are quantitatively quite comparable except for very high levels of noise. We compared PEWA to the baseline NL-means [2] and DCT [26] (using the implementation of www.ipol.im) since they form the core of PEWA. The PSNR values increases of 1.5 db and 1.35 db on average over NL-means and DCT respectively. Finally, we compared the results to the recent S-PLE method which uses SURE to guide the probabilistic patch-based filtering described in [24]. Figure 2 shows the denoising results on the noisy *Valdemossa* ($\sigma = 15$), *Man* ($\sigma = 20$) and *Castle* ($\sigma = 25$) images denoised with BM3D, NL-Bayes and PEWA. Visual quality of methods is comparable.

Table 3 presents the denoising results with PEWA if the pre-computed estimators are obtained with a Wiener filtering (spatial domain[1]) and DCT-based transform [26]. The results of PEWA with $5 \times 5$ or $7 \times 7$ patches are also given in Table 3, for one and two iterations. Note that NL-means can be considered as a special case of the proposed method in which the original noisy patches constitute the set of "weak" estimators. The MCMC-based procedure can be then considered as an alternative procedure to the usual implementation of NL-means to accelerate summation. Accordingly, in Table 3 we added a fair comparison ($7 \times 7$ patches) with the implementation of NL-means algorithm (IPOL (ipol.im)) which restricts the search of similar patches in a neighborhood of $21 \times 21$ pixels. In these experiments, "PEWA basic" (1 iteration) produced better results especially for $\sigma \geq 10$. Finally we compared these results with the most popular and competitive methods on the same images. The PSNR values are selected from publications cited in the literature. LSSC and BM3D are the most

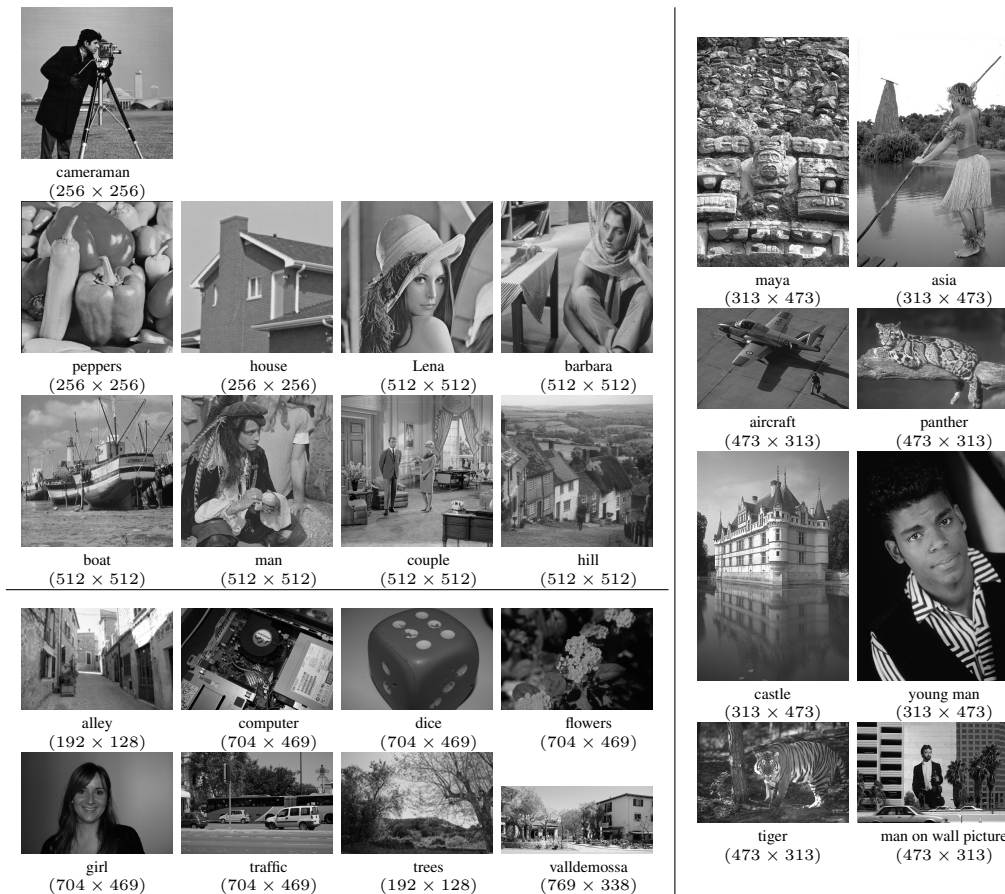

Figure 1: Set of 25 tested images. Top left: images from the BM3D website (cs.tut.fi/˜foi/GCF-BM3D/) ; Bottom left: images from IPOL (ipol.im); Right: images from the Berkeley segmentation database (eecs.berkeley.edu/Research/Projects/CS/ vision/bsds/).

performant but PEWA is able to produce better results on several piecewise smooth images while BM3D is more appropriate for textured images.

In terms of computational complexity, denoising a $512 \times 512$ grayscale image with an unoptimized implementation of our method in C++ take about 2 mins (Intel Core i7 64-bit CPU 2.4 Ghz). Recently, PEWA has been implemented in parallel since every patch can be processed independently and the computational times become a few seconds.

## 6   Conclusion

We presented a new general two-step denoising algorithm based on non-local image statistics and patch repetition, that combines ideas from the popular NL-means [6] and BM3D algorithms [6] and theoretical results from the statistical literature on Exponentially Weighted Aggregation [7, 21]. The first step of PEWA involves the computation of denoised images obtained with a separate collection of multiple denoisers (Wiener, DCT... ) applied to the input image. In the second step, the set of denoised image patches are selectively exploited to compute an aggregated estimator. We showed that the estimator can be computed in reasonable time using a Monte-Carlo Markov Chain (MCMC) sampling procedure. If we consider DCT-based transform [6] in the first step, the results are comparable in average to the state-of-the-art results. The PEWA method generalizes the NL-means algorithm in some sense but share also common features with BM3D (e.g. DCT transform, two-stage collaborative filtering). tches, contrary to NL-Bayes and BM3D. For future work, wavelet-based transform, multiple image patch sizes, robust statistics and sparse priors will be investigated to improve the results of the flexible PEWA method.

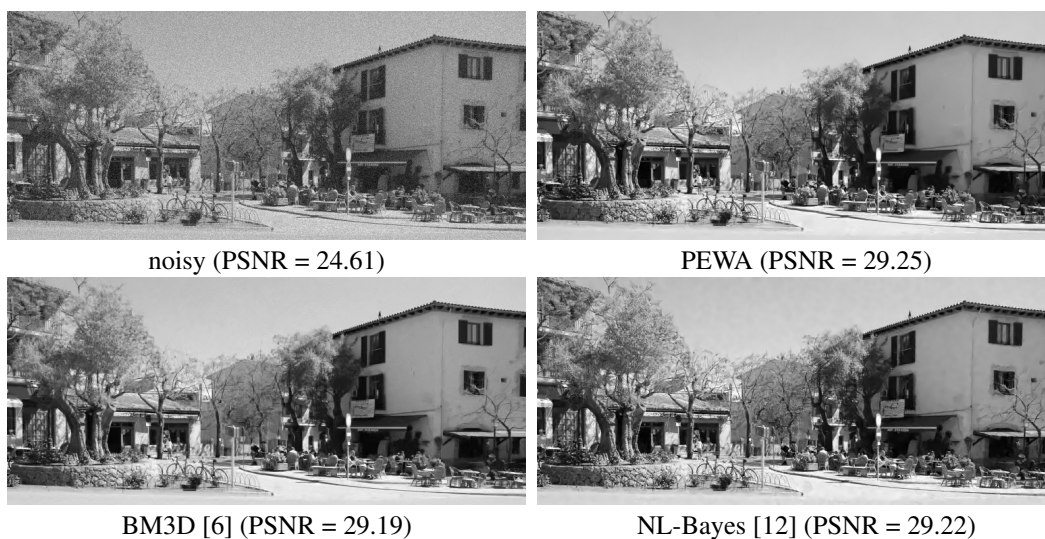

noisy (PSNR = 24.61)     PEWA (PSNR = 29.25)

BM3D [6] (PSNR = 29.19)     NL-Bayes [12] (PSNR = 29.22)

Figure 2: Comparison of algorithms. *Valldemossa* image corrupted with white Gaussian noise ($\sigma = 15$). The PSNR values of the three images denoised with DCT-based transform [26] are combined with PEWA are 27.78, 27.04 and 26.26.)

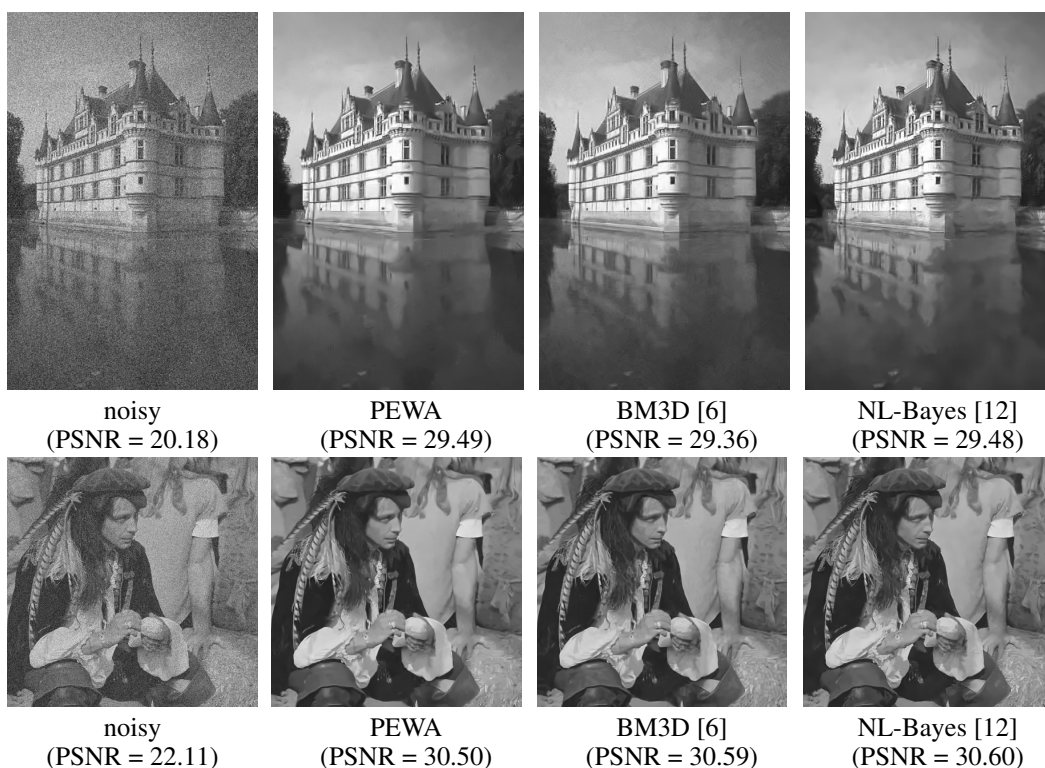

noisy (PSNR = 20.18)   PEWA (PSNR = 29.49)   BM3D [6] (PSNR = 29.36)   NL-Bayes [12] (PSNR = 29.48)

noisy (PSNR = 22.11)   PEWA (PSNR = 30.50)   BM3D [6] (PSNR = 30.59)   NL-Bayes [12] (PSNR = 30.60)

Figure 3: Comparison of algorithms. First row: *Castle* image corrupted with white Gaussian noise ($\sigma = 25$). The PSNR values of the three images denoised with DCT-based transform [26] and combined with PEWA are 25.77, 24.26 and 22.85. Second row: *Man* image corrupted with white Gaussian noise ($\sigma = 20$). The PSNR values of the three images denoised with DCT-based transform [26] and combined with PEWA are 27.42, 26.00 and 24.67.

| | $\sigma = 5$ | $\sigma = 10$ | $\sigma = 15$ | $\sigma = 20$ | $\sigma = 25$ | $\sigma = 50$ | $\sigma = 100$ |
|---|---|---|---|---|---|---|---|
| Cameraman | 38.20 | 34.23 | 31.98 | 30.60 | 29.48 | 26.25 | 22.81 |
| Peppers | 38.00 | 34.68 | 32.75 | 31.40 | 30.30 | 26.69 | 22.84 |
| House | 39.56 | 36.40 | 34.86 | 33.72 | 32.77 | 29.29 | 25.35 |
| Lena | 38.57 | 35.78 | 34.12 | 32.90 | 31.89 | 28.83 | 25.65 |
| Barbara | 38.09 | 34.73 | 32.86 | 31.43 | 30.28 | 26.58 | 22.95 |
| Boat | 37.12 | 33.75 | 31.94 | 30.64 | 29.65 | 26.64 | 23.63 |
| Man | 37.68 | 33.93 | 31.93 | 30.50 | 29.50 | 26.67 | 24.15 |
| Couple | 37.35 | 33.91 | 31.98 | 30.57 | 29.48 | 26.02 | 23.27 |
| Hill | 37.01 | 33.52 | 31.69 | 30.50 | 29.56 | 26.92 | 24.49 |
| Alley | 36.29 | 32.20 | 29.98 | 28.54 | 27.46 | 24.13 | 21.37 |
| Computer | 39.04 | 35.13 | 32.81 | 31.23 | 30.01 | 26.38 | 23.27 |
| Dice | 46.82 | 43.87 | 42.05 | 40.58 | 39.36 | 35.33 | 30.82 |
| Flowers | 43.48 | 39.67 | 37.47 | 35.90 | 34.55 | 30.81 | 27.53 |
| Girl | 43.95 | 41.22 | 39.52 | 38.27 | 37.33 | 34.14 | 30.50 |
| Traffic | 37.85 | 33.54 | 31.13 | 29.58 | 28.48 | 25.50 | 22.90 |
| Trees | 34.88 | 29.93 | 27.49 | 25.86 | 24.69 | 21.78 | 20.03 |
| Valldemossa | 36.65 | 31.79 | 29.25 | 27.59 | 26.37 | 23.18 | 20.71 |
| Aircraft | 37.59 | 34.62 | 33.00 | 31.75 | 30.72 | 27.68 | 24.99 |
| Asia | 38.67 | 34.46 | 32.25 | 30.73 | 29.60 | 26.63 | 24.32 |
| Castle | 38.06 | 34.13 | 32.02 | 30.56 | 29.49 | 26.15 | 23.09 |
| Man Picture | 37.78 | 33.58 | 31.27 | 29.73 | 28.44 | 24.65 | 21.50 |
| Maya | 34.72 | 29.64 | 27.17 | 25.42 | 24.28 | 22.85 | 18.17 |
| Panther | 38.53 | 33.91 | 31.56 | 30.02 | 28.83 | 25.59 | 22.75 |
| Tiger | 36.92 | 32.85 | 30.63 | 29.13 | 27.99 | 24.63 | 21.90 |
| Young man | 40.79 | 37.36 | 35.58 | 34.30 | 33.25 | 29.59 | 25.20 |
| **Average** | **38.54** | **34.75** | **32.67** | **31.26** | **30.15** | **26.95** | **23.76** |

Table 1: Denoising results on the 25 tested images for several values of $\sigma$. The PSNR values are averaged over 3 experiments corresponding to 3 different noise realizations.

| | $\sigma = 5$ | $\sigma = 10$ | $\sigma = 15$ | $\sigma = 20$ | $\sigma = 25$ | $\sigma = 50$ | $\sigma = 100$ |
|---|---|---|---|---|---|---|---|
| PEWA 1 | 38.27 | 34.39 | 32.26 | 30.76 | 29.62 | 26.00 | 22.35 |
| PEWA 2 | 38.54 | **34.75** | **32.67** | **31.26** | **30.15** | **26.95** | 23.76 |
| BM3D [6] | **38.64** | **34.78** | **32.68** | **31.25** | **30.19** | **26.97** | **24.08** |
| NL-Bayes [12] | **38.60** | **34.75** | 32.48 | **31.22** | 30.12 | 26.90 | 23.65 |
| S-PLE [24] | 38.17 | 34.38 | 32.35 | 30.67 | 29.77 | 26.46 | 23.21 |
| NL-means [2] | 37.44 | 33.35 | 31.00 | 30.16 | 28.96 | 25.53 | 22.29 |
| DCT [26] | 37.81 | 33.57 | 31.87 | 29.95 | 28.97 | 25.91 | 23.08 |

Table 2: Average of denoising results over the 25 tested images for several values of $\sigma$. The experiments with NL-Bayes [12], S-PLE[24], NL-means [2] and DCT [26] have been performed using the using the implementation of IPOL (ipol.im). The best PSNR values are in bold.

| Image | Peppers (256 × 256) | | | | House (256 × 256) | | | | Lena (512 × 512) | | | | Barbara (512 × 512) | | | |
|---|---|---|---|---|---|---|---|---|---|---|---|---|---|---|---|---|
| $\sigma$ | 5.00 | 15.00 | 25.00 | 50.00 | 5.00 | 15.00 | 25.00 | 50.00 | 5.00 | 15.00 | 25.00 | 50.00 | 5.00 | 15.00 | 25.00 | 50.00 |
| PEWA 1 (W) (5×5) | 36.69 | 30.58 | 27.50 | 22.85 | 37.89 | 31.88 | 28.55 | 23.49 | 37.27 | 31.43 | 28.30 | 23.45 | 36.39 | 30.18 | 29.31 | 22.71 |
| PEWA 2 (W) (5×5) | 37.45 | 32.20 | 29.72 | 26.09 | 38.98 | 34.27 | 32.13 | 28.35 | 38.05 | 33.40 | 31.11 | 27.80 | 37.13 | 31.94 | 29.47 | 25.58 |
| PEWA 1 (W) (7×7) | 36.72 | 30.60 | 27.60 | 22.82 | 37.90 | 31.90 | 28.59 | 23.52 | 37.26 | 31.45 | 28.33 | 23.45 | 36.40 | 30.18 | 27.32 | 22.71 |
| PEWA 2 (W) (7×7) | 37.34 | 32.34 | 30.11 | 26.53 | 39.00 | 34.57 | 32.51 | 29.04 | 38.00 | 33.65 | 31.56 | 28.40 | 37.00 | 32.10 | 30.00 | 26.20 |
| PEWA 1 (D) (5×5) | 37.70 | 32.45 | 29.83 | 26.01 | 39.28 | 34.23 | 31.79 | 27.72 | 38.46 | 33.72 | 31.33 | 27.59 | 37.71 | 32.20 | 29.55 | 25.58 |
| PEWA 2 (D) (5×5) | 37.95 | **32.80** | 30.20 | **26.66** | 39.46 | 34.74 | 31.67 | 29.15 | 38.57 | 33.96 | 31.81 | 28.43 | 38.03 | 32.70 | 30.03 | 26.01 |
| PEWA 1 (D) (7×7) | 37.71 | 32.43 | 29.87 | 26.00 | 39.27 | 34.26 | 31.79 | 27.71 | 38.45 | 33.72 | 31.25 | 27.62 | 37.70 | 32.30 | 29.84 | 26.20 |
| PEWA 2 (D) (7×7) | 38.00 | 32.75 | **30.30** | 26.69 | 39.56 | 34.83 | 32.77 | 29.29 | 38.58 | 34.12 | 31.89 | 28.83 | 38.09 | 32.86 | 30.28 | 26.58 |
| PEWA Basic (7×7) | 36.88 | 31.34 | 29.47 | 26.02 | 37.88 | 34.13 | 32.14 | 28.25 | 37.39 | 33.26 | 31.20 | 27.92 | 36.80 | 31.89 | 29.76 | 25.83 |
| NL-means [2] (7×7) | 36.77 | 30.93 | 28.76 | 24.24 | 37.75 | 32.36 | 31.11 | 27.54 | 36.65 | 32.00 | 30.45 | 27.32 | 36.79 | 30.65 | 28.99 | 25.63 |
| BM3D [6] | 38.12 | 32.70 | 30.16 | **26.68** | 39.83 | 34.94 | 32.86 | 29.69 | **38.72** | **34.27** | **32.08** | **29.05** | 38.31 | **33.11** | **30.72** | **27.23** |
| NL-Bayes [12] | 38.09 | 32.26 | 29.79 | 26.10 | 39.39 | 33.77 | 31.36 | 27.62 | **38.75** | 33.51 | 31.16 | 27.62 | 38.38 | 32.47 | 30.02 | 26.45 |
| ND-SAFIR [11] | 37.34 | 32.13 | 29.73 | 25.29 | 37.62 | 34.08 | 32.22 | 28.67 | 37.91 | 33.70 | 31.73 | 28.38 | 37.12 | 31.80 | 29.24 | 24.09 |
| K-SVD [10] | 37.80 | 32.23 | 29.81 | 26.24 | 39.33 | 34.19 | 31.97 | 28.01 | 38.63 | 33.76 | 31.35 | 27.85 | 38.08 | 32.33 | 29.54 | 25.43 |
| LSSC [16] | **38.18** | **32.82** | 30.21 | 26.62 | **39.93** | **35.35** | **33.15** | **30.04** | 38.69 | 34.15 | 31.87 | 28.87 | **38.48** | 33.00 | 30.47 | 27.06 |
| PLOW [5] | 37.69 | 31.82 | 29.53 | 26.32 | 39.52 | 34.72 | 32.70 | 29.08 | 38.66 | 33.90 | 31.92 | 28.32 | 37.98 | 21.17 | 30.20 | 26.29 |
| SOP [18] | 37.63 | 32.40 | 30.01 | 26.75 | 38.76 | 34.35 | 32.54 | 29.64 | 38.31 | 33.84 | 31.80 | 28.96 | 37.74 | 32.65 | 30.37 | 27.35 |

Table 3: Comparison of several versions of PEWA (W (Wiener), D (DCT), Basic) and competitive methods on a few standard images corrupted with white Gaussian noise. The best PSNR values are in bold (PSNR values from publications cited in the literature).

## Footnotes

[1] $u_\ell(x) = \text{mean}(v(x)) + \max\left(0, \dfrac{\text{var}(v(x)) - a_\ell\sigma^2}{\text{var}(v(x))}\right) \times (v(x) - \text{mean}(v(x)))$, where $\ell = \{1, 2, 3\}$ and $a_1 = 0.15, a_2 = 0.20, a_3 = 0.25$.

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
