[Reviews · NeurIPS 2014]

Submitted by Assigned_Reviewer_12

This paper proposes a denoising algorithm based on non-local image statistics and patch repetition by combining the advantages of NL-means and Exponentially Weighted Aggregation (EWA). The computation of the aggregated estimator is done using MCMC and results are comparable to state-of-the-art algorithms.

Pluses:
1) the method seems simple and straightforward to implement.
2) the experiments and results are convincing.

Minus:
In terms of explaining *why* the method works, the text leaves something to be desired. E.g., in the second paragraph of section 7 ("The proposed implementation proceeds in two identical iterations.") there is no explanation or motivation, besides that "it works".
To be clear, most other parts of the method are somewhat better justified than the example I just gave, but in many cases the reader is left to guess why this or that choice was made or why it improved the results.

In general, the text can be improved quite a bit if more effort would be put into providing
additional explanations.

Another example:
``..., several patches are over-represented in the average and many patches are not selected.'' This is confusing: is the over-representation (or under-representation of other patches) a good thing or a bad thing? The mere fact that it is sparse (as mentioned in the next line) doesn't answer this question. May the authors please elaborate on this in their rebuttal?

Minor remarks:
In reference [2], the year has been omitted (2005?).
Line 083: ``enables''--> ``enable''.
Line 182: ``sens''--> ``sense''
Line 230: ``equal weight'' --> ``equal weights''.
Line 307-308: In this sentence, the word ``recently'' should probably appear only once, not twice.
Summary: The method is interesting and sufficiently novel (even though it is not ground-breaking), and the results are convincing.
The authors could have improved the way in which they explain and motivate some of their choices and steps. Should the paper get accepted, I encourage them
to try to improve this important issue. Having said that, overall my general impression is still positive.

Submitted by Assigned_Reviewer_13

The paper proposes a new patch-based image denoising algorithm.
The paper combines ideas from the popular Nonlocal-Means algorithm and
theoretical results from the statistics literature on the SURE risk and
exponentially weighted aggregation of multiple estimators.

The proposed algorithm denoises each patch separately by the following
procedure: An initial denoising algorithm is first applied (here DCT with
$L$ different settings) to produce denoised versions of the image. The set
of all overlapping patches in these seed images (together with the raw noisy
image) supplies a large number of "weak" estimators. The proposed algorithm
aggregates these "weak" estimators to produce an estimate of the clean image
values. The patches are aggregated with SURE-derived exponential weights (also
complemented by a spatial proximity term).

NL-means can be considered a special case of the proposed method in which the
original noisy patches constitute the set of "weak" estimators.

The paper proposes an MCMC-based procedure to accelerate summation. It would
be interesting to compare the proposed MCMC method with the acceleration
heuristics used in the NL-means algorithm (confine search to a small local
window, fast ANN search etc). It would be nice to see this issue investigated
if the paper gets accepted.

The authors do a good job evaluating the proposed method. It performs better
than NL-means and similarly to BM3D (which is considered to be the
state-of-art in image denoising and a fast implementation is available).

Small typo:
049: EPPL -> EPLL
Summary: In my opinion, the paper is worth publishing. It proposes a relatively novel
method and nicely puts it in context of prior work, brings some new
perspective in the area, and the proposed algorithm performs around the
state-of-art.

Submitted by Assigned_Reviewer_41

This paper proposes a general aggregation method for image denoising by combining multiple weakly denoised images from standard methods. The proposed method is based on EWA and it employs Monte-Carlo simulation for efficient approximation. The results are comparable to the state-of-the-art algorithms.

It is nice for the authors to provide a thorough review on the image denoising literature and to discusses the difference of PEWA with existing denoising algorithms in the beginning. As far as I understand, the main contributions of this paper (Sec 4) compared to other "internal" methods are that PEWA generate candidate patches from weakly denoised images and that it merges candidate patches based on EWA. The aggregation method is flexible enough to combine any standard denoising algorithm and has a nice interpretation with Gibbs distribution. The experiments show that the aggregated image can be significantly better than each component, and good results can be obtained efficiently.

I have a concern on the flexibility of the proposed method. The authors criticize the "external" methods for not being flexible for unknown noise level in line 55-57. But PEWA also relies on a known noise level. What if the noise variance is unknown? And what if the noise is not iid Gaussian?

I'm also worried about the fact that when combining a large number of denoised images, the quality will drop (line 244-246). Would that suggest the SURE is not good enough to combine a large number of patches, or the MCMC approach is not mixing well?

In the experiment section, PEWA is shown to perform just similar to and sometimes worse than the state-of-the-art. It would be useful to discuss any advantage of PEWA compared to BM3D and NL-Bayes.
Summary: The proposed method is flexible enough to aggregate the results of any denoising algorithm and shows results comparable to the state-of-the-art. However, it requires a known noise level, and its advantage against the state-of-the-art algorithms is not clear.
Author Feedback
Author rebuttal: We thank the reviewers for their relevant comments. The reviewers’ criticisms are mainly concerned with the reasons why the PEWA method is able to produce results comparable to state-of-the-art algorithms. Actually, the paper equally presents the theory and the experimental results/illustrations. Due to the lack of space, it was hard to meet these two objectives and the paper outlines the main features of PEWA, without addressing all concerns raised by the reviewers as it would be necessary.

Reviewer #12:
In the rebuttal, we try to justify the choices we have made, as asked by the reviewer.

Most of state-of-the-art algorithms (e.g. BM3D, NL-Bayes) use two iterations. The idea is to exploit the first iteration to help the algorithm to improve the overall result at the second iteration. For instance, NL-Bayes updates the covariance matrices of groups of patches and computes an improved estimator at the second iteration. In our approach, the denoised image obtained at the first iteration is potentially over-smoothed at some location. At the second iteration, PEWA combines the “first” patch estimators and the “weak” patch estimators to better restore the over-smoothed structures. For low signal-to-noise ratios, PEWA considers the “first” patch estimators as “good” new candidates and the unwanted residual noise of the first iteration is removed by the second averaging process.

PEWA exploits several denoised versions of the input image to adapt to a variety of local contexts. It is generally a hard task to fix a unique threshold (DCT, Wavelet Transform…) to denoise satisfyingly an image since important structural details are irremediably lost. Considering a set of weak denoised images allows us to overcome this difficulty and to preserve important structures slightly drowned in the noise.

In the MCMC sampling, some patches are more frequently selected than others at a given location. The number of occurrences of a particular candidate patch can be evaluated. In constant image areas, there is probably no preference for any one patch over any other but we expect to select a low number of candidate patches along image contours and discontinuities. The adjective “sparse” was maybe a misleading choice but we meant that the method is able to use a very small set of candidate patches for restoration at some location, depending on image contexts.

In this paper, we will try to better justify the other choices we have made while preserving the description of experimental results.

Reviewer #13:
The MCMC procedure may be considered as an efficient computational method to calculate the NL-means estimator as noticed by the reviewer. Nevertheless, it has been shown that considering all the image patches to restore a given pixel does not produce the best results (except for repeated textures) and it is computationally demanding. It is more recommended to exploit patches taken a semi-local neighborhood (e.g. 21 x 21 search windows) to get higher PSNR values. In our approach, the neighborhood is not fixed but the range is a priori controlled by the parameter \tau. It is then possible to select relevant patches located far from the central patch (not taken in a fixed neighborhood). As suggested, we can add a fair comparison with NL-means by considering only noisy patches to constitute the set of “weak” estimators. This can be useful to demonstrate that combining several denoised images definitely plays a key role in PEWA.

Reviewer #41:
For real-world noisy images, we use the method described in Kervrann and Boulanger (IEEE T. Image Processing 2006) to robustly estimate the noise variance. Generally, state-of-the-art methods are based on a Gaussian noise model and generally provide very satisfying results in real applications. This Gaussian approximation is then considered as sufficient in most cases. For signal-dependent noise, we suggest to compute spatially adaptive noise variances combined with PEWA. This was successfully experimented but not shown in this paper. Finally, the SURE estimator can be appropriately designed to remove specific noise models. For instance, PURE was recommended in Luisier et al. (Signal Processing 2010) to remove Poisson noise.

In line 244-246, we mentioned that considering a large number L of denoised drops quality. In our experiments, the results look very similar for an unchanged number of MCMC iterations (T=1000). Increasing L would suggest to considering more MCMC samples since the space of candidate patches is higher.

The reviewer also noticed that SURE could be not good enough to combine a larger number of denoised images. Actually, we assume that all estimators are independent in PEWA since it is unfeasible to compute SURE as defined by Stein. However, investigating other robust influence functions (e.g. Huber, Leclerc…), similar to \phi(z) = |z| (see (4)), in the definition of the Gibbs energy could be envisaged.

Unlike BM3D and NL-means, combining several “weak” estimators is an important feature of PEWA. This is not considered in BM3D even if Wavelet Transform or PCA have been investigated instead of DCT. Actually, the components of BM3D are well known in signal processing (DCT, Wiener filtering) but the combination proposed by Dabov et al. is under study yet since it is hard to do a better job of denoising. NL-Bayes computes the covariance matrices and the mean of groups of similar patches. It shares common features with BM3D in some sense and both algorithms use two iterations. Our method shares more common features with NL-means, which is a more intuitive algorithm. PEWA does not explicitly group similar patches, contrary to NL-Bayes and BM3D.

We provide a more comprehensive approach than NL-Bayes and BM3D without sacrificing efficiency. The PEWA framework can mix, on a well-founded statistical basis, several estimators coming from different methods (Wiener, DCT...) and robust statistics can be investigated.